# Low potential enzymatic hydride transfer via highly cooperative and inversely functionalized flavin cofactors

Max Willistein[1], Dominique F. Bechtel[2], Christina S. Müller[3], Ulrike Demmer[4], Larissa Heimann [3], Kanwal Kayastha [4], Volker Schünemann[3], Antonio J. Pierik [2], G. Matthias Ullmann [5], Ulrich Ermler[4] & Matthias Boll [1]

Hydride transfers play a crucial role in a multitude of biological redox reactions and are mediated by flavin, deazaflavin or nicotinamide adenine dinucleotide cofactors at standard redox potentials ranging from 0 to –340 mV. 2-Naphthoyl-CoA reductase, a key enzyme of oxygen-independent bacterial naphthalene degradation, uses a low-potential one-electron donor for the two-electron dearomatization of its substrate below the redox limit of known biological hydride transfer processes at $E^{o\prime} = -493$ mV. Here we demonstrate by X-ray structural analyses, QM/MM computational studies, and multiple spectroscopy/activity based titrations that highly cooperative electron transfer ($n = 3$) from a low-potential one-electron (FAD) to a two-electron (FMN) transferring flavin cofactor is the key to overcome the resonance stabilized aromatic system by hydride transfer in a highly hydrophobic pocket. The results evidence how the protein environment inversely functionalizes two flavins to switch from low-potential one-electron to hydride transfer at the thermodynamic limit of flavin redox chemistry.

[1] Microbiology, Faculty of Biology, University of Freiburg, Schänzlestrasse 1, 79104 Freiburg, Germany. [2] Biochemistry, Faculty of Chemistry, University of Kaiserslautern, Erwin-Schrödinger-Straße 52, 67663 Kaiserslautern, Germany. [3] Biophysics, Department of Physics, University of Kaiserslautern, Erwin-Schrödinger-Straße 46, 67663 Kaiserslautern, Germany. [4] Max-Planck-Institute for Biophysics Frankfurt, Max-von-Laue-Str. 3, 60438 Frankfurt, Germany. [5] Computational Biochemistry, University of Bayreuth, Universitätsstrasse 30, NW I, 95447 Bayreuth, Germany. Correspondence and requests for materials should be addressed to U.E. (email: ulrich.ermler@biophys.mpg.de) or to M.B. (email: matthias.boll@biologie.uni-freiburg.de)

The biodegradation of aromatic compounds to $CO_2$ by aerobic and anaerobic microorganisms is important for the global carbon cycle and for the elimination of persistent aromatic pollutants. In particular polycyclic aromatic hydrocarbons (PAH)s are classified as harmful for the environment and human health[1,2]. While aerobes employ oxygenases to attack aromatic ring systems, such a strategy is no option for anaerobic bacteria. Here, enzymatic dearomatization is typically accomplished by reduction affording electron transfer at the negative redox potential limit in biology[3–6].

In the anaerobic degradation pathways of most monocyclic aromatic compounds, the key intermediate benzoyl-coenzyme A (CoA) serves as substrate for ATP or electron-bifurcation dependent benzoyl-CoA reductases[7–9]. They dearomatize their substrate to a cyclic, conjugated 1,5-dienoyl-CoA at $E°' = -622$ mV[10] in one-electron steps using either a [4Fe–4S] cluster or a tungstopterin cofactor as ultimate one-electron donors. A radical-based reaction mechanism has been proposed for enzymatic benzoyl-CoA dearomatization[11]. Remarkably, the analogous chemical Birch reduction proceeds at harsh conditions involving cryogenic temperatures, alkali metals in ammonia as reducing agents and alcohols as proton donors[12].

During anaerobic degradation of the PAH model compound naphthalene, 2-naphthoyl-CoA reductase (NCR) reduces 2-naphthoyl-CoA (NCoA) to 5,6-dihydro-2-naphthoyl-CoA (DHNCoA), (Fig. 1)[13–15]. To date only the NCR from the sulfate-respiring enrichment culture N47 has been studied to some extent. It belongs to a distinct subclass of the old yellow enzyme (OYE) family of flavoproteins that are composed of three domains binding FMN (flavin mononucleotide), a [4Fe–4S] cluster and either ADP or flavin adenine dinucleotide (FAD)[13]. Recent studies showed that NCoA reduction in $D_2O$ enantioselectively yielded the product with the (5S, 6S)-configuration[16]. The observed defluorination of 6-F-2-NCoA to 2-NCoA by NCR is in agreement with a hydride transfer to C6 followed by protonation of the thioester-stabilized intermediate at C5 (Fig. 1)[16]. However, the redox potential of the NCoA/5,6-DHNCoA couple is with $E°' = -493$ mV[13] outside the range of known biological hydride carriers such as NAD(P)H, flavins or deazaflavins (F420) with $E'°$ values ranging between ≈0 and $-340$ mV[17–22]. In contrast, biological electron transfer processes below $-400$ mV usually involve one-electron carriers comprising metal cofactors. In addition, FMN in flavodoxins can act as low-potential one-electron carriers ($E°' \approx -420$ mV) that switch between stable neutral semiquinone (SQ) and instable anionic hydroquinone (HQ) states[23,24]. Notably, for NCR only artificial low-potential one-electron donors such as sodium dithionite or Ti(III)–citrate, but not NAD(P)H served as donors. A low-potential ferredoxin or a ferredoxin-like domain of an oxidoreductase serves most likely as natural electron donor[14].

Taken these observations together, the mechanism of enzymatic naphthoyl-ring reduction has remained enigmatic. The two alternative mechanistic scenarios for naphthoyl ring dearomatization involve either a hydride transfer far below the redox window of known biological hydride-transfer reactions, or the transfer of single electrons or hydrogen atoms in a Birch reduction like manner via radical intermediates. In any scenario, catalysis at the proposed active site FMN cofactor would involve previously unnoticed flavin redox chemistry. Here, we address this question by solving the X-ray structure of NCR with and without substrate combined with integrative electrochemical, spectroscopic and computational analyses. The results demonstrate how the protein environment functionalizes two flavin cofactors as low-potential one- and two-electron carriers that allow for hydride transfer to an aromatic ring system at the negative limit of the biological redox scale.

## Results

**Overall structure of NCR.** The X-ray structure of NCR was determined at 2.2 Å resolution with $R/R_{free}$ values (%) of 18.9/21.9 (for complete statistics of crystal structure analysis see Table 1). NCR is present in the crystal as a monomer modularly built up of a OYE-like TIM-barrel domain (19–359) hosting the active site FMN, an FAD binding α/β domain (401–503 + 626–674), and a second α/β domain (504–625). The [4Fe–4S] cluster is harbored in a linker region (360–400) inserted between the TIM barrel domain and the FAD binding domain (Fig. 2). This three-domain architecture is also present in the structurally characterized 2,4-dienoyl-CoA reductase (DCR, 2-*trans*-enoyl-CoA forming using NADPH as electron donor)[25] as well as trimethylamine[26] and histamine dehydrogenases[27] (using electron-transferring flavoprotein [ETF] as electron acceptor). The rms deviation between NCR and DCR from *E. coli* is 1.7 Å (93% of residues used) at a sequence identity of 33% (1PS9)[25], and that between NCR and OYE acting on non-CoA ester substrates is ca. 1.8 Å (around 53% of residues used) at a sequence identity of around 27% (e.g., 3KRU[28], 3HF3[29], and 4UTK[30]).

**Cofactor binding and electron transfer to the active site.** The architecture of NCR clearly reveals a spatial separation between electron uptake at the FAD near the surface and substrate reduction at the deeply buried active site FMN cofactor.

The two flavins are electronically connected via a [4Fe–4S] cluster with edge to edge distances of 6 Å to FMN and 9 Å to FAD, respectively (Fig. 2). The FAD cofactor of NCR is located at the C-terminal end of the central β-sheet with its isoalloxazine moiety clamped between L456, W459, and L656 on the *si*-side and E541, I542, and K655 on the *re*-side (Supplementary Fig. 1b). The surrounding of the isoalloxazine by negatively charged residues (E446, E541, and D649) destabilize the FADH⁻ (deprotonated FAD HQ), but not the FADH• (protonated FAD SQ) state suggesting a low-redox potential, similar to that of flavins in flavodoxins (Fig. 3).

The amide side chain of Q545 and the hydrophobic L456 and W459 point to the N5 of FAD and preferably stabilize the oxidized FAD state (Fig. 3). A distinguishing feature between DCR and NCR is the presence of an NADPH binding site in DCR which is absent in NCR. This finding can be rationalized by the exceptionally low potential of the NCoA/DHNCoA couple ($E°' = -493$ mV), which is ≥250 mV lower than for standard substrates of OYEs and far too negative for NADPH ($E' \approx -360$ mV) as reductant. The bulky W459 in NCR (F424 in DCR) shifts the isoalloxazine moiety towards the *re*-side and thus totally blocks the NADPH binding site present in DCR (Supplementary Fig. 1).

**Fig. 1** Stereochemical course and potential hydride transfer mechanism of NCR. The enantioselective hydride (blue) and proton (red) transfers are shown

**Table 1 Data collection and refinement statistics**

| | NCR | NCR-NCoA (soaked) | NCR-DHNCoA (co-cryst) |
|---|---|---|---|
| **Data collection** | | | |
| Space group | $P2_1$ | $P2_1$ | $I4_1$ |
| Resolution (Å) | 50–2.2 | 50–2.2 | 50–2.4 |
| Cell dimensions | | | |
| $a, b, c$ (Å) | 82.5, 86.8, 96.9 | 81.7, 86.1, 96.9 | 176.9, 176.9, 49.2 |
| $\alpha, \beta, \gamma$ (°) | 90.0, 90.7, 90.0 | 90.0, 90.7, 90.0 | 90.0, 90.0, 90.0 |
| $R_{sym}$ (%) | 7.5 (102.2)[a] | 4.9 (100.9) | 7.1 (95.2) |
| $I / \sigma I$ | 13.2 (1.8) | 9.2 (1.0) | 12.9 (1.2) |
| Completeness (%) | 99.5 (99.7) | 95.8 (91.1) | 99.1 (99.8) |
| Redundancy | 4.0 (4.1) | 2.0 (2.0) | 5.7 (6.1) |
| B-factor (Wilson plot) | 42.0 | 49.8 | 59.0 |
| **Refinement** | | | |
| Resolution (Å) | 50.0–2.2 | 50.0–2.2 | 50.0–2.4 |
| No. of reflections | 69,124 | 67,470 | 29,943 |
| $R_{work}/R_{free}$ | 18.9/ 21.9 | 18.2 / 22.4 | 23.0/27.3 |
| No. of atoms | | | |
| Protein | 10,191 | 10,168 | 5057 |
| Ligands/ion | 184 | 304 | 152 |
| Water | 187 | 242 | 27 |
| B-factors | | | |
| Protein | 52.0 | 62.8 | 93.3 |
| FAD, [4Fe-4S], FMN, substrate | 42.1, 52.6, 37.2 | 49.7, 61.5, 46.0, 89.7 | 103.8, 71.7, 62.7, 90.5 |
| Water | 43.5 | 58.4 | 76.8 |
| R.m.s. deviations | | | |
| Bond lengths (Å) | 0.015 | 0.015 | 0.004 |
| Bond angles (°) | 1.70 | 1.72 | 0.91 |

[a]Values in parenthesis are for highest-resolution shell. Each dataset is based on one crystal

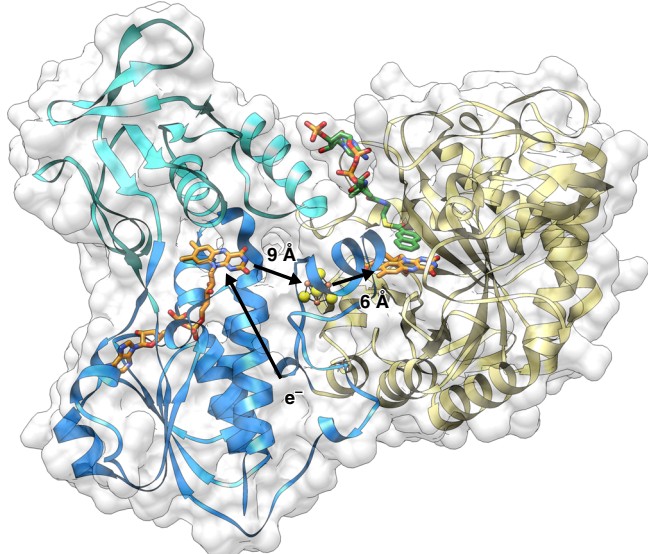

**Fig. 2** Overall X-ray structure. NCR was found in the crystal structure as a monomeric enzyme in agreement with previous gel filtration data. It is composed of an OYE-like TIM barrel domain (yellow) and by two α/β domains (blue/cyan). The bound cofactors and NCoA (carbons in green) are shown as balls and sticks. A ferredoxin or ferredoxin-like domain of an oxidoreductase has been suggested as natural external electron donor; its proposed binding site is indicated by e⁻/arrow

Notably, trimethylamine or histamine dehydrogenases bind ADP instead of FAD; here, electrons are directly shuttled via the [4Fe–4S] cluster to the ETF acceptor[26,27].

A ferredoxin or a ferredoxin-like domain of an oxidoreductase ($E' \approx -500$ mV) is assumed to serve as in vivo electron donor system for NCR[14]. Its postulated binding site is located in a flat hollow between the two α/β domains lined up by residues N540, I542, D569, M626, V653, and K655. The distance between a potential Fe/S cluster from the donor to the FAD is estimated to be below 14 Å.

The Fe/S cluster in NCR was found in the crystal structure as a cysteine-coordinated [4Fe–4S] cluster, albeit with an incomplete occupancy. Mössbauer spectroscopic analyses with $^{57}$Fe-labeled oxidized NCR revealed spectra that were in a typical range for $Fe^{2.5+}$ ions present in diamagnetic $[4Fe–4S]^{2+}$ clusters. Mössbauer spectra of the dithionite reduced state were characteristic for $Fe^{2.5+}$ and $Fe^{2+}$ pairs of a reduced $[4Fe–4S]^{1+}$ cluster[31]. These results indicate the presence of an electron-transferring $[4Fe–4S]^{2+/1+}$ cluster (for Mössbauer parameters and spectra see Supplementary Tables 1 and 2, and Supplementary Fig. 3).

The conformation and binding site of FMN is basically conserved among members of the OYE family. Thus, N5 of the isoalloxazine ring is linked with the polypeptide in all family members by a main chain amine hydrogen bond donor stabilizing the oxidized FMN state. However, two major differences between NCR and standard OYEs were identified in the polypeptide surrounding of the isoalloxazine ring that should substantially modify its electrostatic/hydropathic properties (Fig. 4). (i) The helices 371:380 and 572:585 shield the dimethylbenzene ring from the bulk solvent in NCR; both are absent in all standard OYEs. In NCR, they build up a unique phenylalanine cluster (F375, F379, and F573) that together with the bicyclic naphthoyl-ring of the substrate form a highly hydrophobic patch (Fig. 4). (ii) In standard OYEs, two histidines and one arginine surround the pyrimidine ring of FMN thereby stabilizing anionic intermediates. From these three only the arginine is present in NCR (R245), but a hydrogen bond to the negatively charged E341 counteracts its positive charge contacting the N1–C2–O group of FMN. E341 is absent in DCR, which instead contains an extra positively charged histidine (H26) in the vicinity

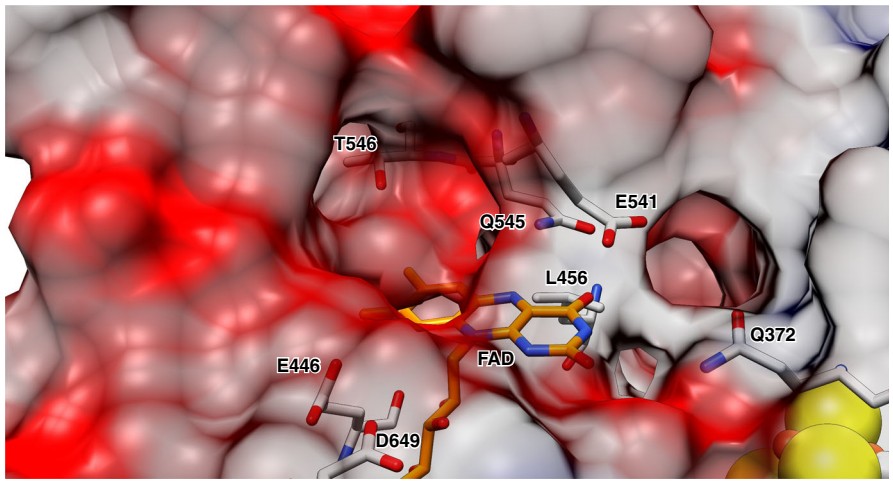

**Fig. 3** Binding of the FAD cofactor. Anionic/polar residues responsible for the negative electrostatic potential are indicated. Several hydrophobic side chains around the isoalloxazine preferably stabilize the uncharged FAD and FADH• states

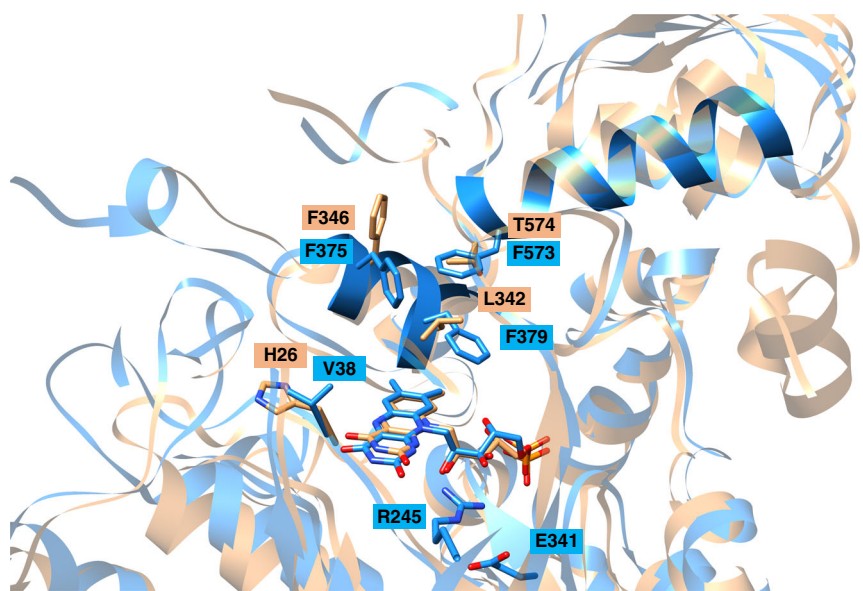

**Fig. 4** Binding of the active site FMN cofactor. Overlay of NCR (blue) and DCR (amber). The FMN is shielded from bulk solvent by the two highlighted helices 371:380 and 572:585 that harbor F375, F379, and F573 creating a hydrophobic pocket absent in other OYEs. The hydride-donating N5 of the FMN is in close proximity of H26 in DCR, which is replaced by a hydrophobic V38 in NCR

of FMN. Theses structural features largely contribute to a stabilization of the oxidized FMN state and thus to a substantial decrease of its redox potential. Electrostatic calculations gave a p$K$a of 8.8 for FMN indicating that $FMNH^-$ rather than $FMNH_2$ is the catalytically relevant reduced state (pH optimum is 6.8).

**Binding of NCoA/DHNCoA**. The NCoA structure in complex with the substrate NCoA and the product DHNCoA were determined at 2.2 and 2.4 Å resolutions, respectively. No significant conformational changes were found between the two binary NCR complex structures and between them and the substrate-/product-free NCR as documented in rms deviations of 0.45 and 0.49 Å, respectively.

NCoA is embedded into a preformed approximately 20 Å deep cavity formed by three β-barrel strands of the TIM barrel domain, the following loops, helix 371:380 of the linker region and helix 572:585 of the second α/β domain. Its "S"-like shape is due to a kink at the cysteamine moiety and to two 90° kinks before and after the ribose of CoA (Fig. 5). The adenine base becomes thereby

attached to the pantetheine moiety. In this conformation, the phospho-ADP moiety of CoA serves as a plug to lock the entrance of the cavity. Multiple, mostly van-der-Waals contacts are formed between residues of the cavity and NCoA (Supplementary Fig. 4), which rationalizes the apparent low $K_m$ of 1.1 μM[14]. The aromatic rings of NCoA are bound in an encapsulated hydrophobic pocket formed by Y82, H84, Y166, I194, V195, F375, V378, F379, and the isoalloxazine ring of FMN. The planar naphthoyl and isoalloxazine rings are oriented almost parallel to each other with the *re*-side stacked over the *si*-side, respectively. Strong π–π stacking interactions are formed between the equidistant distal phenyl and pyrazine ring atoms. Only little space is left at the cavity bottom of NCR beyond the distal phenyl ring of the substrate. This observation is in agreement with the ability of NCR to convert 6-F-NCoA but not the bulkier 2-phenanthroyl-CoA[14,16].

**Redox titrations**. NCR (20 μM) was stepwise titrated with sodium dithionite at pH 8.0; at this pH the redox potential of

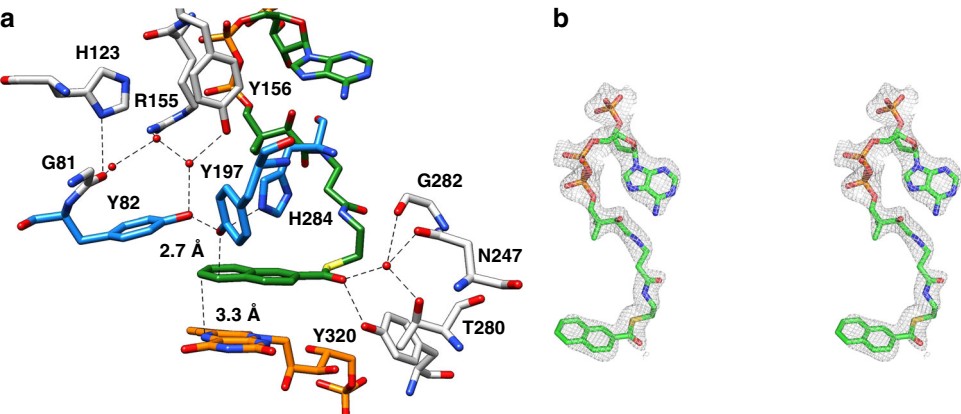

**Fig. 5** Binding site of NCoA. **a** Active site geometry. The distances of hydride transfer from N5 of FMN (orange) and proton transfer from Y197 to the naphthoyl ring are indicated (carbons are shown in green). Y197 is hydrogen bonded with H284 and Y82 (all highlighted in blue), both contributing to an enhanced acidity; the proton network to the bulk solvent via bound water molecules involves Y156, R155, G81, and H123. A potential enolate intermediate at the thioester carbonyl during NCoA reduction is stabilized by Y320, G282, N247, and T280. **b** Stereo image of NCoA with the corresponding omit map (as gray meshes at a contour level of $0.5\sigma$). NCoA is bound in an 'S'-like conformation. Although not completely occupied, all moieties of NCoA are also clearly visible in the $2F_o$–$F_c$ electron density map at a contour level of $1\sigma$

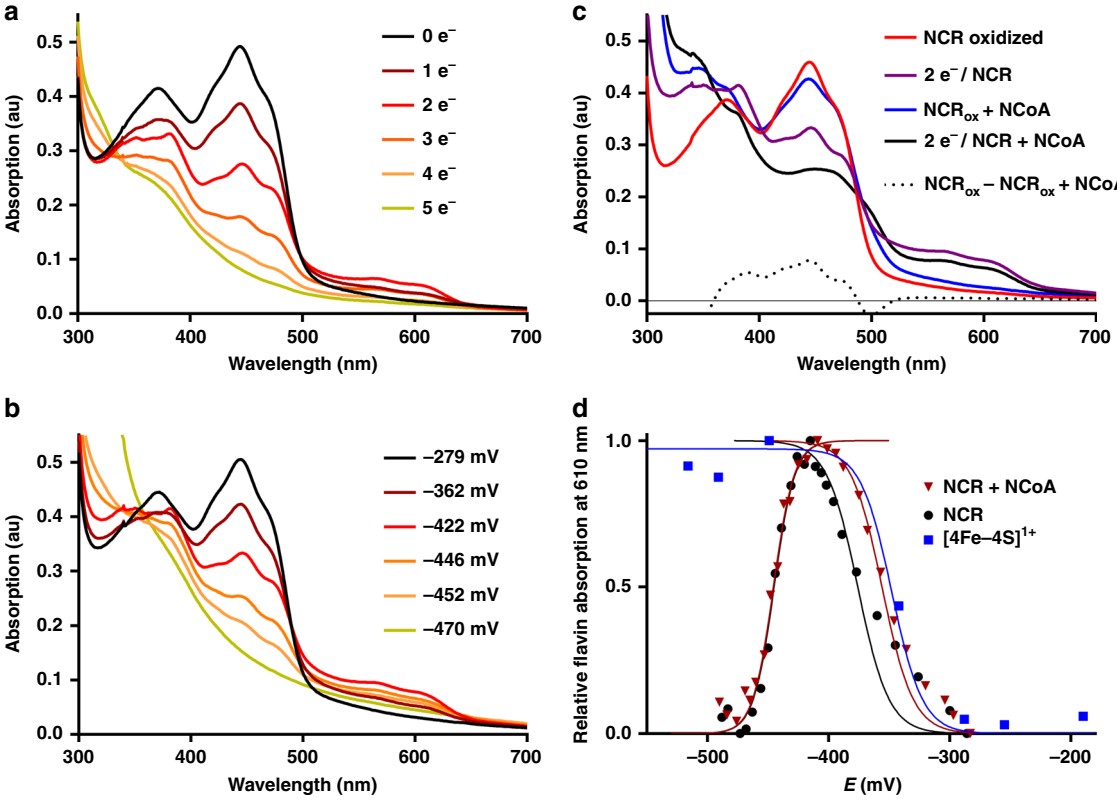

**Fig. 6** Titrations of NCR (20 μM) with dithionite at pH 8. **a** Absorption spectra after addition of stoichiometric electron equivalents. **b** Absorption spectra at defined redox potentials. **c** Absorption spectra of oxidized and two-electron reduced NCR in the absence and presence of NCoA. In the latter the signal of a charge-transfer complex between the substrate and FMN remained that allowed assigning the SQ formation to the FAD cofactor, whereas FMN remained fully oxidized. The dotted line represents the difference spectrum of oxidized NCR without NCoA minus oxidized NCR with NCoA. **d** Titration of the neutral SQ absorbance spectra of FAD in the absence (black) and presence (red) of NCoA. Formation of SQ/reduction of [4Fe-4S] was fitted to a Nernst curve with $n = 2$ electrons; reduction of SQ with $n = 3$ due to strong cooperativity of parallel FADH• and FMN reduction. Titration of [4Fe-4S] cluster was followed by its $S = 1/2$ EPR spectrum (blue). Symbols represent experimental data, solid lines are fitted curves. $R^2 = 0.98$ (+NCoA)/0.78 for SQ formation; $R^2 = 0.98$ (+NCoA)/0.99 for SQ reduction. au absorbance units. Source data are provided as a Source Data file

dithionite is below –550 mV allowing for a complete reduction of all cofactors. Flavin reduction was monitored by recording UV/ vis spectra; reduction of the [4Fe–4S] cluster was followed by EPR spectroscopy. Redox potentials during titrations were determined

using an Ag/AgCl reference electrode and redox-mediating dyes. This setup allowed the assignment of the number of reducing equivalents added and of the redox potentials poised to the redox states of cofactors (Fig. 6a, b). The oxidized absorption spectrum

of NCR was characteristic for neutral FMN and FAD cofactors with maxima at 372 and 444 nm as reported earlier[13].

In titrations without mediators/reference electrode, full reduction of 10 µM NCR was achieved with 25 µM of the two-electron donor dithionite, which perfectly fits to the five-electron reduction of the FMN, FAD, and [4Fe–4S]$^{2+}$ cofactors. Upon stepwise reduction, the spectrum of the oxidized flavins gradually bleached between −280 and −470 mV vs. SHE. The characteristic spectrum of a neutral SQ appeared with absorption maxima at 566 nm and 610 nm. It optimally developed after addition of two-electron equivalents at ≈−420 mV and disappeared upon further reduction. Formation and disappearance of the SQ spectrum followed Nernst curves giving $E^0{}' = -353 \pm 2$ mV (quinone[Q]/SQ), and $E^0{}' = -446 \pm 1$ mV (SQ/HQ), respectively (Fig. 6d). Surprisingly, best fits were obtained for a two-electron transition ($n = 2$) for the Q/SQ couple and a three-electron transition ($n = 3$) for the SQ/HQ couple (Supplementary Fig. 5). This finding suggests a strong cooperative behavior of the three NCR cofactors in terms of joint co-reduction at highly similar apparent potentials (see below). Such cooperativity has previously only been described for the two cofactors of cytochrome $cd_1$[32].

In the oxidized state, only a very weak $g = 2.02$ [3Fe–4S]$^{1+}$ EPR signal as substoichiometric breakdown product of the [4Fe–4S]$^{2+}$ was detected, in full agreement with Mössbauer spectroscopic data (Supplementary Fig. 3). During the reductive, dye-mediated titration, the rise of a broad rhombic $S = 1/2$ EPR signal with g values of 2.08, 1.91, and 1.8 developed and remained stable at decreasing potentials (Supplementary Fig. 6). Its formation followed a Nernst curve with $E^0{}' = -348 \pm 10$ mV with $n = 2$ being very close to that observed for the Q/SQ couple (Fig. 6d). These results indicate that addition of two-electron equivalents reduced one flavin to the SQ and the [4Fe–4S] cluster to its +1 state in a highly cooperative manner at highly similar potentials.

In the presence of NCoA, the absorbance spectrum of oxidized NCR was clearly affected by the formation of a shoulder around 500 nm (Fig. 6c). This broadening is assigned to a charge-transfer complex between the substrate and the active site FMN and represents a distinguishing feature of FMN compared to FAD. After reduction of NCR by two electrons in the presence of NCoA, the remaining UV/vis spectrum was dominated by the charge-transfer complex signal, whereas the features of the oxidized flavin spectrum in the absence of NCoA were almost completely lost. Moreover, the redox potentials of flavin SQ formation and reduction were only marginally affected by NCoA (Fig. 6d). Both findings clearly indicate that the observed SQ-signal derives from FAD and not from FMN and corroborate the proposed cooperative reduction of FAD to FADH• and the [4Fe–4S]$^{2+}$ to the 1+ state upon addition of two-electron equivalents; in contrast the FMN remained fully oxidized.

Upon further reduction of the NCR–NCoA complex, at potentials below −420 mV, the spectra of the FADH• and the FMN/NCoA charge-transfer complex (at 444 nm, Supplementary Fig. 7) both simultaneously decreased. The redox potential dependent reduction of FMN to FMNH$^-$ with virtually no intermediary SQ state fitted best to a Nernst curve with $E^0{}' = -439 \pm 1$ mV and $n = 3$ (Supplementary Fig. 7). These values are very close to those obtained for FADH• reduction ($E^0{}' = -446 \pm 1$ with $n = 3$) and indicate a strong cooperativity between the one-electron reduction of FADH• and the two-electron-reduction of FMN to FMNH$^-$. When NCoA was omitted from the redox titration assay, the reduction of FMN also followed a Nernst curve with an only slightly more negative $E^0{}' = -445 \pm 5$ mV with $n = 3$ (Supplementary Fig. 7). Reduction of NCR by excess of the two-electron donor DHNCoA (exergonic reverse reaction) was incomplete, and the neutral flavin SQ

remained close to maximum intensity (Supplementary Fig. 8). Together with results obtained in titrations with dithionite, this finding indicates that FMN readily accepts a hydride from DHNCoA, and the FMNH$^-$ formed reduces the FAD and the [4Fe–4S]$^{2+}$ cluster by single electrons, each.

The dependence of the extent of NCoA reduction by NCR on the redox potential poised was determined by analyzing samples by ultra performance liquid chromatography (Fig. 7a). At potentials ≥−420 mV (two-electron reduced NCR) no conversion of NCoA was observed using established assays. Upon further reduction, the extent of NCoA conversion continuously increased and was maximal when NCR was completely reduced. The reduction of maximally 20% of the NCoA added at equal amounts to NCR can be explained by reaching thermodynamic equilibrium because $E^0{}'$ of the NCoA/DHNCoA couple ($E^0{}' = -493$ mV) is around 25 mV more negative than fully reduced NCR (at $E' \approx -470$ mV). Notably, the increase of NCoA reduction extent at potentials below −400 mV almost perfectly correlated with the decrease of the FADH• spectrum suggesting that only the fully reduced FADH$^-$ is competent of supplying FMN via the [4Fe–4S] cluster with single electrons for hydride transfer to NCoA.

The results obtained from titration experiments and their interpretation is summarized in Fig. 7; a corresponding electro-chemical landscape for the electron transfer events is presented in Fig. 8. They suggest that the three redox cofactors transfer electrons in a highly cooperative manner as indicated by the almost identical redox potentials of the FAD/FADH• and [4Fe–4S]$^{2+/1+}$ ($E^0{}' \approx -350$ mV) as well as the FADH•/FADH$^-$ and FMN/FMNH$^-$ ($E^0{}' \approx -445$ mV) couples. Only at redox potentials below −420 mV, NCR gradually becomes competent for NCoA reduction accompanied by the cooperative ($n = 3$) reduction of the FADH• to FADH$^-$ and the FMN to FMNH$^-$. The absence of any UV/vis feature of a flavin SQ signal at low potential suggests that the FMNH• is extremely instable with $E^0{}'(Q/SQ) \ll E^0{}'(SQ/HQ)$. Such highly differing crossed-over redox potentials of the first and second electron transfer are essential for two-electron donor systems[33].

**Mechanism of NCR reaction.** One of the major aims of this work was to substantiate or falsify the proposed low-potential hydride transfer mechanism. Results from redox titrations already indicated a highly instable FMNH• that is immediately further reduced to FMNH$^-$, which argues for a two-electron hydride transfer process. In the NCR–NCoA complex structure, the orientation of the aromatic rings and the distance between the N5 of FMN and the C6 of NCoA of 3.3 Å are optimal for transferring a hydride. Its trajectory was simulated using quantum-mechanical (QM)/MM calculations by starting from the reduced anionic FMNH$^-$ state. Using PyCPR[34], we found a transition state at 16 kcal mol$^{-1}$ in which the proton of FMNH$^-$ is at half distance between N1 of FMN and C6 of NCoA, and the HOMO is delocalized over both ring systems (Fig. 9). Moreover, an intermediate with an energy of 4.5 kcal mol$^{-1}$ (relative to the initial state) is formed in which the hydride from FMN is already fully transferred to NCoA, but Y197 still holds its proton. In the calculation, no radical intermediate was identified, in contrast to an earlier study on the mechanism of benzoyl-CoA reduction for which a similar computational setup (conjugate peak refinement with BP86) was used[35]. The negative charge of the transition state is delocalized over the conjugated ring system including the thioester (Fig. 1). The transition state is stabilized by hydrogen-bond interactions between the carbonyl oxygen and the Y320 hydroxy group and a polypeptide-linked water molecule (Fig. 5 and Supplementary Fig. 4). In NCR, Y197 is properly placed at the

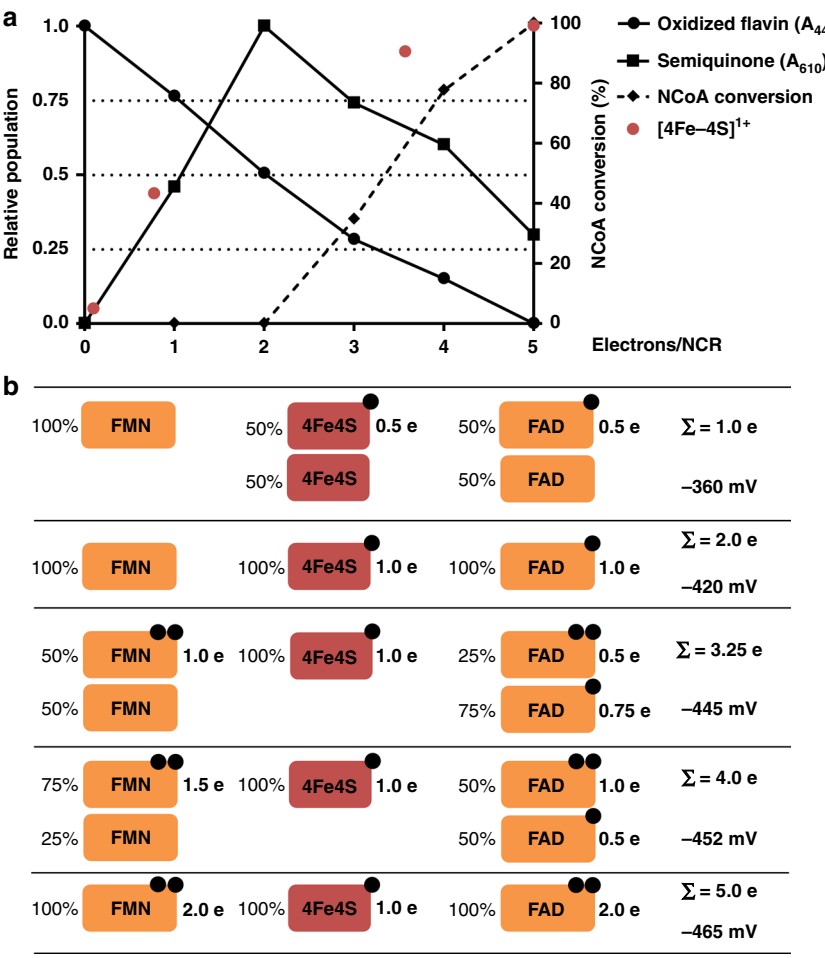

**Fig. 7** Summary of redox titration experiments. **a** Relative populations of oxidized flavins, neutral SQ, [4Fe–4S]$^{1+}$ cluster and extent of NCoA reduction after addition of up to five electron equivalents to oxidized NCR. Species were determined by UV/vis spectroscopy at 444 nm (oxidized flavins), and 610 nm (neutral SQ); [4Fe–4S]$^{1+}$ cluster by EPR spectroscopy. **b** Cartoon summarizing reduction of individual NCR cofactors depending on the number of electron equivalents added/redox potential. Source data are provided as a Source Data file

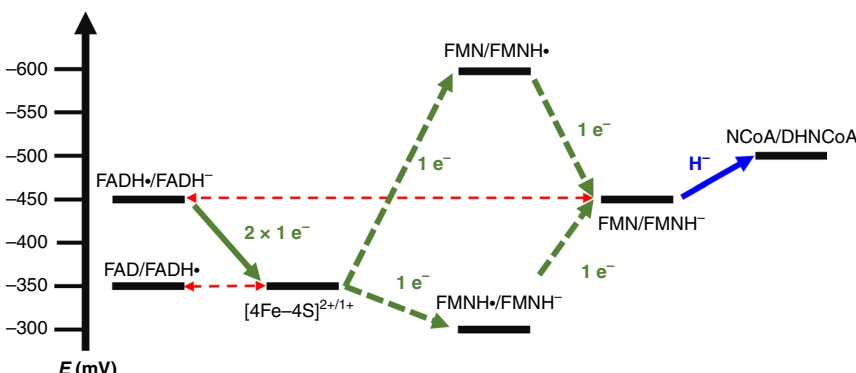

**Fig. 8** Electrochemical landscape for the electron transfer reactions during NCR catalysis. After one-electron reduction of the [4Fe–4S] cluster and FAD, respectively, two-single electrons are transferred to FMN to form a hydride after protonation that is subsequently transferred to NCoA. While FAD switches only between the SQ/HQ states, an FMN SQ intermediate was never observed. This finding is rationalized by largely differing crossed-over redox potentials of the two redox transitions of FMN (the values shown are minimal estimates based on the inability to observe the FMN SQ in the steady state). As a result, the second reduction drives the unfavorable first reduction resulting in the observed two-electron reduction of FMN. For the NCoA/DHNCoA couple the standard redox potential is given, which under cellular conditions will be slightly more positive

*si*-side of the substrate with a distance between its oxygen and C5 of 2.7 Å strongly suggesting a function as proton donor during NCoA reduction. According to QM/MM calculations Y197 transfers the proton to C5 of the 6-hydro-NCoA anion with a small barrier of only about 2 kcal/mol.

The invariant proton-donating tyrosine[36] is activated in all OYE members by a single interaction with a hydrogen-bond donor such as tyrosine, histidine or asparagine. NCR appears to be the only known OYE family member where the acidity of the proton-donating Y197 is increased by interactions with the two

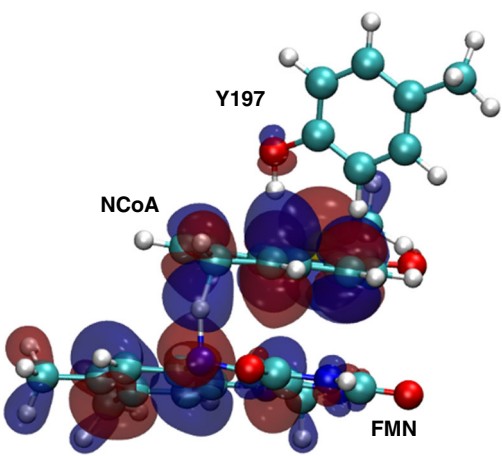

**Fig. 9** Highest occupied molecular orbital (HOMO) of the transition state for the hydride transfer from FMN to NCoA. The HOMO is delocalized over both ring systems and the proton is located between N5 of FMN and C6 of NCoA. Y197 is ready to transfer its proton to the reduced NCoA

hydrogen-bond donors H284 and Y82 (Fig. 5). We calculated the protonation pattern of the active site residues at pH 7 using continuum electrostatic methods. The $pK_a$ value of Y197 at pH = 7 is 12.6[37], which is low enough to donate a proton to a reduced anionic intermediate of NCoA. Its surprisingly high value is due to the desolvation effect in a rather hydrophobic active site pocket which is compensated by the local electrostatic field. H284 seems to be uncharged due to the hydrophobic surrounding and its role as proton acceptor in the hydrogen bond to the cysteamine N–H of CoA. The deprotonated state of H284 (calculated $pK_a$ of −4.1 at pH = 7) cannot reprotonate Y197. Y82 (calculated $pK_a$ of 16.0 at pH = 7) may donate a proton to Y197 supported by the finding that its hydroxyl group also serves as the ending point for a proton channel from bulk solvent. The proton channel extends via a water molecule bound to Y156 and two further water molecules that are hydrogen-bonded to H123 and R155 to bulk solvent (Fig. 5).

In summary, the presented structural data provide the complete molecular basis for the regio- and stereoselectivity of NCR reaction via hydride and proton transfer events. They are in full agreement with data obtained by previous vibrational circular dichroism spectroscopy analyses[16].

## Discussion

The presented structural, spectroscopic, and computational analyses strongly support a hydride transfer mechanism for NCR at a redox potential far below previously reported values for flavin dependent (de)hydrogenases. Consequently, the previous opinion that reversible biological electron transfer below the standard redox potential of canonical organic redox cofactors is restricted to metal cofactors or one-electron-transferring flavodoxins appears not to be valid.

The most challenging aspects of NCR catalysis is the generation of the exceptionally low-potential FMNH⁻ hydride donor by a low-potential single-electron donor system. For this purpose two electronically coupled flavins are functionalized by the polypeptide surrounding in an opposite manner to low-potential one-electron (FAD) and hydride (FMN) carriers. The almost identical one-electron (FAD SQ/HQ) and two-electron (FMN/FMNH⁻) redox potentials result in the observed highly cooperative behavior ($n = 3$) of the two flavins, which is essential for the low-potential one-/two-electron transfer switch. An equilibration of single electrons over the two flavin cofactors is suppressed by the crossed-over single-electron redox potential of FMN with the

$E°'$(Q/SQ) $\ll$ $E°'$(SQ/HQ)[33,38], which impedes reoxidation of the FMNH⁻ formed to FMNH•. The highly unfavorable first one-electron reduction of FMN ($E°' \ll$ −500 mV) by FADH⁻ via the [4Fe–4S]$^{2+/1+}$ cluster will be greatly pulled forward by the highly exergonic second electron transfer ($E°' \gg$ −400 mV) (Fig. 8). Most likely, the generation of the FMNH⁻ state represents the rate-limiting step of the slow NCR reaction (40 nmol mg⁻¹ min⁻¹)[13]. Similar short-living flavin SQ states generated by a low-potential one-electron donor were proposed and identified in flavin-based electron-confurcation processes. There, the second electron for reduction to the HQ state originates from a spatially separated high-potential one-electron donor[39–41].

The question rises whether the approximately $E°' = −450$ mV for the FMN/FMNH⁻ couple described in this work marks the negative redox limit of enzymatic hydride transfer? Considering the lowest one-electron redox-potential for a flavodoxin FMN of around −500 mV, the border of a hydride transfer may be also anticipated in this range. At least for the even more challenging reduction of benzoyl-CoA ($E°' = −622$ mV) flavin mediated hydride transfer appears to be no option as benzoyl-CoA reductases employ a one-electron reduction strategy using metals as active site cofactors.

## Methods

**Expression and purification of NCR.** NCR was heterologously produced as strep-tag fusion protein in *E. coli* using autoinduction medium inoculated with a fresh overnight culture. Cells were grown at 37 °C for four hours followed by incubation at 16 °C for 20 h. Cells were disrupted by a single passage through a french pressure cell at 1100 psi after resuspension in 2 mL of buffer A (50 mM HEPES pH 8.0, 150 mM) per g cells (wet weight). Cell debris was removed by centrifugation at 100,000$g$ for 60 min and the supernatant was applied to streptactin affinity resin (IBA Lifesciences, Göttingen, Germany) equilibrated with buffer A using an ÄKTA purifier system (GE Healthcare, Solingen, Germany). Unbound proteins were washed out using buffer A and NCR was eluted with 5 mM D-desthiobioton in buffer A[16]. For Mössbauer spectroscopy 200 μM of $^{57}$Fe(III)citrate were added to the growth medium instead of nonlabeled iron source. For crystallization, peak fractions after affinity chromatography were pooled and diluted with buffer H$_A$ (20 mM HEPES/KOH pH 8.0) to a concentration of 50 mM KCl or lower. Then the protein solution was loaded onto a Resource Q column (6 mL, GE Healthcare, Solingen, Germany) pre-equilibrated with buffer H$_A$. The column was washed with 6% (60 mM KCl) of buffer H$_B$ (20 mM HEPES/KOH pH 8.0, 1 M KCl) before bound proteins were eluted in a linear gradient of 6-10% H$_B$ (60–100 mM KCl). Elution fractions were pooled and concentrated and further purified by gel filtration using a Superdex 200 PG 26/600 column (320 mL, GE Healthcare, Solingen, Germany) using 20 mM HEPES/KOH pH 7.6 with 100 mM KCl. Peak fractions were pooled and concentrated to 17–30 mg/mL and frozen in liquid nitrogen and stored at −80 °C until further use (for sodium dodecyl sulfate polyacrylamide gel electrophoresis see Supplementary Fig. 9).

**Redox titrations coupled to UV/vis spectroscopy.** UV/vis spectroscopy was performed in an anaerobic chamber at 25 °C using a spectrophotometer (UV-1650PC, Shimadzu, Duisburg, Germany) and quartz cuvettes. All buffer and enzyme solutions were rendered anaerobic using N$_2$ before the experiments. All spectra were normalized for their absorption at 700 nm. For titration with sodium dithionite, NCR was diluted to a concentration of 10 μM in assay buffer (100 mM TRIS/HCl pH 8.0) and titrated in 2 μM steps using a freshly prepared stock solution of sodium dithionite (0.4 mM in assay buffer). For better comparison, the resulting spectra were extrapolated to a protein concentration of 20 μM. For determination of redox potentials, NCR was diluted to a concentration of 20 μM in buffer A (50 mM HEPES pH 8.0, 150 mM KCl) supplemented with a mix of redox mediators (Supplementary Table 3) at a concentration of 0.1 μM each to a final volume of 2 mL[42]. The solution was constantly stirred in a fluorescence cuvette and the potential was measured using a Ag/AgCl redox electrode (InLab Redox Micro, Mettler-Toledo, Giessen, Germany) calibrated with saturated quinhydrone solutions at pH 7.0 and 4.0. All values were obtained from single determinations, and were corrected to potentials versus H$_2$/H$^+$ using +207 mV as potential for the Ag/AgCl reference electrode. NCR was incubated for 30 min before titration with sodium dithionite in buffer A and UV/vis spectra were recorded at defined redox potentials after allowing the potential to stabilize ($\Delta E < 2$ mV/min). For titration in the presence of substrate, NCoA was added to a final concentration of 50 μM. Data were analyzed and depicted using Prism 6 (GraphPad, San Diego, USA). For fitting of Nernst curves to data points, a modified Nernst equation was used[43].

**Mössbauer and EPR spectroscopy.** Mössbauer and EPR spectroscopic analyses are described in the Supplementary methods.

**Crystallization and structure determination**. Crystallization of NCR was performed with the vapor diffusion method at 4 and 20 °C. The protein solution contained 25 mg ml$^{-1}$ NCR, 20 mM HEPES, pH 7.5, 0.5 mM FAD and 0.5 mM FMN. The drop solution consists of 2 µL of protein solution, 1 µL of precipitate solution (composed of 17% PEG 4000 or 20% PEG3350), 0.18–0.2 M NaF and 1 µL silver bullet solution A12 or BioD4[44]. The corresponding reservoir solution and 20% (w/v) glycerol served for cryo-protection. Data were collected at the Swiss-Light-Source, Villigen at a wavelength of 1 Å and processed with XDS[45] (Table 1). Phases were determined by molecular replacement using PHASER[46] integrated into PHENIX[47]. A suitable model was prepared by taking DCR of *E. coli* (1PS9) without residues 517–553 and by partially pruned side chains using the phenix_sculptor script (option Schwarzenbacher)[48]. After refining the obtained model with REFMAC[49] and PHENIX, ARP/WARP[50] was used for amino acid exchange and further refinement. After completing manual model building with COOT[51] refinement was terminated with BUSTER (Phaser, Global Phasing Ltd.) using NCS and TLS parameters[52]. Moreover, NCR crystals were soaked with 0.5 mM NCoA (pH 6.5–7.0) for 2 h at 4 °C and frozen. The X-ray structure was refined with BUSTER using the coordinates of the substrate-free NCR as starting model. Although not completely occupied, all component parts of NCoA were clearly visible in the 2$F_o$-$F_c$ electron density map at a contour level of 1σ. The composite omit map at a contour level of 0.5σ is shown in Fig. 5b. Co-crystallization between NCR and NCoA (5 mM) or DHNCoA (5 mM) was performed in an anaerobic tent (95% N$_2$, 5% H$_2$) to detect the influence of O$_2$ and crystal lattice effects for substrate/product binding. Crystals essentially grew under the same conditions (20% PEG 3350, 0.2 M NaF) but adopted a new crystal form. Refinement of the NCR–DHNCoA complex structure was performed with PHENIX. The obtained model is highly flexible and partly disordered (overall B-factor: 93.1 Å$^2$); the B-factor around DHNCoA is with ca. 65 Å$^2$ significantly lower. In the three data sets are 95–97% of the backbone dihedral angle in favored regions and 0.0–0.2% in outlier regions.

**Computational modeling**. As starting model for calculations, the structure of NCR with bound NCoA substrate was used. A sphere of water with a radius of 30 Å centered at the active site of NCR was added to the system to represent the solvent. The structural model was prepared using the software CHARMM and together with its force field[53] analogous to the procedure in a previous publication[35]. The FeS cluster was assumed to be reduced and its charges were taken from the literature[54]. The force field parameters for NCoA were taken from analogous parameter sets in the CHARMM force field. The assignment of the protonation state of the titratable side chains was based on titration calculations using a continuum electrostatic model and Monte Carlo titrations[55,56]. In the QM/MM calculations, the following residues comprised the QM region: the isoalloxazine ring of FMN, the naphthoyl-thioester part of NCoA, and side chains of Y82, H123, Y197, H284 and three active site water molecules bridging the gap between H123 and Y82. All side chains were truncated between Cα and Cβ. In the active site, Y82 and Y197 were set to their protonated state and H284 was set to its neutral form (protonated at Nε2). For FMN, the negatively charged reduced state was assumed (deprotonated N1). The QM/MM calculations were performed using pDynamo[57] together with ORCA[58]. We used unrestricted DFT as QM method, namely we used the BP86 functional with the def2-SVP basis set for the search of the path and recalculated the energies using M06 with a def2-TZVP basis. The MM energies were calculated using the CHARMM27[59] force field. To model the QM/MM boundary, a link-atom scheme and electrostatic embedding was used. The QM region was surrounded by a fully flexible MM layer of 8 Å. MM atoms that had a distance of more than 8 Å from any QM atom were harmonically restrained. The force constants of the restraints were linearly increased between 8 and 16 Å from 0 to 100 kcal (mol Å)$^{-1}$, except for the iron–sulfur center, which was restrained by a force constant of 100 kcal (mol Å)$^{-1}$. Beyond 16 Å, the maximal force constant 100 kcal/(mol Å) was applied for the restraints. Using a conjugate gradient minimizer, we minimized the initial coordinates. To obtain first estimates for transition and intermediate states, we scanned the potential energy surface along the relevant bonds with a step size of 0.1–0.4 Å. The RMS gradient threshold of 0.04 kcal (mol Å)$^{-1}$ was used for all minimizations and surface scans. For finding reliable transition states, we started from these estimates of the intermediates and transition state and applied the PyCPR[34] implementation of the conjugated peak refinement method[60].

**Reporting summary**. Further information on research design is available in the Nature Research Reporting Summary linked to this article.

## Data availability

The datasets generated and analyzed during the current study are available from the corresponding authors upon reasonable request. The source data underlying Fig. 6a–d, Fig. 7a, b and Supplementary Figs. 3, 5–9 are provided in the Source Data file. In addition, the PDB files for generating Figs. 2–5 and Supplementary Figs. 1, 2 are accessible online at the protein data bank (6QKG, 6QKR, and 6QKX).

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

## Acknowledgements
This work was funded by the German Research Foundation (DFG) within RTG 1976 (M.W. and M.B.) and SPP 1927 (M.B., A.P., V.S., and M.U.). We thank Hartmut Michel for continuous support and the staff of the SLS, Villigen for help in data collection. We thank Sina Weidenweber for initial crystallization.

## Author contributions
M.W. purified enzymes and conducted biochemical, spectroscopic, kinetic analyses. D.F.B., and A.J.P. carried our EPR spectroscopy experiments. U.D., K.K., and U.E. crystallized enzymes and analyzed X-ray structural data, C.S.M., L.H., and V.S. performed the Mössbauer spectroscopic experiments. M.U. conducted the computational studies. M.B. designed the overall project. The paper was written through contributions of all authors. All authors have given approval to the final version of the paper.

## Additional information

**Competing interests:** The authors declare no competing interests.

