## [Peer Review File · Nature Communications]

Reviewers' comments:

Reviewer #1 (Remarks to the Author):

This work offers details that will further our understanding of how anaerobic bacteria function in the degradation of aromatic compounds. The authors present a detailed look at the mechanism by which NCR catalyzes substrate reduction at the limit of flavin redox chemistry. Basing their model in structural studies that are supported with computational work and spectroscopic analyses, their arguments are robust and back up their proposed mechanism. The roles of two flavins and the biochemistry of the enzyme active site in achieving this energetically difficult biosynthetic transformation should be of interest to the biochemical community.

The arguments presented about the active site, including cofactors and important residues, seem generally reasonable. Though probably outside the scope of this narrative, it might be interesting to perform biochemical assays (mutagenesis) to confirm the important and seemingly novel requirement of both H284 and Y82 to increase the acidity of Tyr197. While it is understandable to argue that negatively charged residues (E446, E541, E658) that surround the isoalloxazine moiety of FAD would destabilize FADH⁻, it seems questionable to include K656 as a negatively charged residue (see line 95).

The manuscript as is would likely be interesting to researchers working in this specific area of enzymology. Overall interest and appreciation for the work presented here could be increased for the general readership of the journal by providing some more context for the work, perhaps in the introduction and certainly in the conclusions, offering a clearer understanding of the mechanism presented here and the unusual and boundary-pushing nature of the transformation achieved in the NCR active site. Perhaps moving Fig 8 from the Supporting Information section to the Conclusions section would facilitate broader understanding of the mechanism elucidated on the part of the reader.

In general, the manuscript could benefit from careful reading and copy editing to make sure that awkward phrasing does not get in the way of scientific understanding. For example, the word "donor" may be missing at the end of line 21 in the abstract. Phrasing in lines 30, 62, 154 could use editing, and the units seem to be wrong in reporting the NCR specific activity on line 315. It was not immediately obvious what an "external protein donor" was when reading the legend for Fig 2 (line 90), since it's actually an external electron donor that is proposed to be a protein, so perhaps reword that sentence as well.

Reviewer #3 (Remarks to the Author):

Reduction of 2-naphtoyl-CoA to 5,6 dihydro-2-naphtoyl-CoA by 2-naphtoyl-CoA reductase requires FAD, an iron sulfur cluster and FMN as cofactors. This reaction is below the negative redox limit usually encountered in biological hydride transfer using these cofactors. The paper by Willistein et al addresses the structural and biochemical basis of this unusual flavin redox chemistry by a combination of crystallography, spectroscopy and computational studies. The authors propose a mechanism by which the enzyme generates the extremely low-potential FMNH⁻ donor necessary for the reaction to occur. The paper is an important advance in understanding enzymatic hydride transfer at the limit of redox chemistry using flavins.

Comments:

The authors should include a composite omit map or the initial map after a MR solution was obtained showing the electron density for the bound substrate, NCoA as a separate panel in figure 5.

Based on the PDB validation reports, there appears to be a significant problem with the crystal structure obtained from the co-crystallized enzyme-substrate complex. Not only are the B-values after refinement very high, the RSRZ value, i.e. the fit of the modelled residues to the electron

density is with more than 20% outliers at the lowest end of all structures deposited in the PDB. The authors should address these problem in the manuscript.

B- values for the bound ligands, FAD, FMN and substrate should be included in table 1. It would also be helpful to add the B-factor from the Wilson plot to this table.

There is a significant variation in the number of water molecules modelled, for instance 150 and 250 for structures at the same resolution and only 9 at a slightly lower resolution – why?

PDB accession codes should be added to the manuscript.

In figure 2 of the ms and figure 2 in the supplementary the authors propose the binding site for the the electron donor, a hypothetical low-potential ferredoxin-like protein. How was this site identified and what would be the basis for this proposal? Figure S2 highlights the negative electrostatic potential of this binding site. Would this negative potential not counteract electron transfer from the ferredoxin to FAD?

Page 6, line 116: occupancy, not occupation

Page 19, line 364: Table 1, not Table 3

Reviewer #4 (Remarks to the Author):

This is an interesting paper combining a range of experimental and computational studies designed to throw light on an important enzymatic mechanism. I am not an expert on the enzyme itself and only comment in depth on the computational aspects.

A standard QM/MM protocol has been followed. My major concern is the use of a B3LYP functional for the QM part. Whilst this functional generally predicts quite good structures, it often fails to predict reliable energetics, due in large part to its failure to properly model dispersive interactions which are expected to be important in bio systems. Most current QM/MM studies now include dispersive effects either explicitly by the use of the DFT-D method or implicitly by the use of modern functionals such as the M0 family of Truhlar.

I believe that the calculations should be repeated using one or more of these modern functionals to give confidence in the validity of the predictions.

Ian H Hillier

Point by point response to the reviewers' comments (own comments in blue)

Reviewers' comments:

Reviewer #1 (Remarks to the Author):

This work offers details that will further our understanding of how anaerobic bacteria function in the degradation of aromatic compounds. The authors present a detailed look at the mechanism by which NCR catalyzes substrate reduction at the limit of flavin redox chemistry. Basing their model in structural studies that are supported with computational work and spectroscopic analyses, their arguments are robust and back up their proposed mechanism. The roles of two flavins and the biochemistry of the enzyme active site in achieving this energetically difficult biosynthetic transformation should be of interest to the biochemical community.

The arguments presented about the active site, including cofactors and important residues, seem generally reasonable. Though probably outside the scope of this narrative, it might be interesting to perform biochemical assays (mutagenesis) to confirm the important and seemingly novel requirement of both H284 and Y82 to increase the acidity of Tyr197. While it is understandable to argue that negatively charged residues (E446, E541, E658) that surround the isoalloxazine moiety of FAD would destabilize FADH-, it seems questionable to include K656 as a negatively charged residue (see line 95).

We agree with the reviewer that site-directed mutagenesis would be the logical topic for a follow-up study including the structural, spectroscopic and kinetic characterization of the mutants. However, as the reviewer stated himself, such a study would be out of the scope of the narrative. The paper includes already multiple approaches (X-ray structural, EPR/Mössbauer/UV/vis spectroscopy, electrochemical titrations, QM/MM computational). Additional mutant characterization would really go beyond the scope and the format of this work.

We deleted listing the amino acids (in particular the mentioned K656, that was listed by mistake as a negatively charged one).

The manuscript as is would likely be interesting to researchers working in this specific area of enzymology. Overall interest and appreciation for the work presented here could be increased for the general readership of the journal by providing some more context for the work, perhaps in the introduction and certainly in the conclusions, offering a clearer understanding of the mechanism presented here and the unusual and boundary-pushing nature of the transformation achieved in the NCR active site. Perhaps moving Fig 8 from the Supporting Information section to the Conclusions section would facilitate broader understanding of the mechanism elucidated on the part of the reader.

We agree with the reviewer and shifted Fig S8 from the Supporting Information to the main text. We thoroughly revised the entire conclusion and the entire manuscript with regard to providing more context of the work and the boundary-pushing nature.

In general, the manuscript could benefit from careful reading and copy editing to make sure that awkward phrasing does not get in the way of scientific understanding. For example, the word "donor" may be missing at the end of line 21 in the abstract. Phrasing in lines 30, 62, 154 could use editing, and the units seem to be wrong in reporting the NCR specific activity on line 315. It was not

immediately obvious what an “external protein donor” was when reading the legend for Fig 2 (line 90), since it’s actually an external electron donor that is proposed to be a protein, so perhaps reword that sentence as well.

We agree with all points raised here by the reviewer and made all the appropriate corrections.

Reviewer #2:

General remark: we greatly appreciate the exceptionally detailed review which greatly helped to improve the manuscripts.

The manuscript by Willistein et al. describes X-ray structure analysis and spectroscopic analysis of 2-naphthoyl-CoA reductase (NCR) to figure out the dearomatization of 2-naphthoyl-CoA (NCoA) and redox states of NCR during the catalysis. NCR contains three cofactors, electron accepting FAD domain, [4Fe-4S] cluster and catalytic FMN. The authors clearly demonstrates that hydride ($2e^-$) transfer from the reduced form of FMN dearomatizes NCoA and the two electrons are virtually transferred at once through [4Fe-4S], which can retain only one electron, by three-electron relay. Though the authors address the single turnover of the reaction and further studies expected to be conducted to reveal the whole catalytic cycle, the study reports novel and significant findings, and should be published after minor revision.

Specific points:

1. The manuscript clearly demonstrates that hydride transfer is an exclusive route for dearomatization of NCoA, but the dearomatization itself had been revealed by the authors' group previously. Therefore, a more specific title should be given to the study.

We changed the title that highlights now the major outcome of the story more precisely.

2. Line 47-49: The sentence refers to Figure 1, but fluoride cannot be found in Figure 1. The sentence should be illustrated by a scheme other than Figure 1.

We agree with this point and deleted the statement of the 'nucleophilic substitution of a fluoride by a hydride' as it is indeed not shown. We rather focus here that the observed defluorination at C6 is in line with a hydride transfer to C6.

3. Line 53-55: The sentence is somewhat incomprehensible. The relation between "low" potential of flavodoxin and the stabilization of SQ and destabilization of HQ need to be explained (redox potentials of the species).

We agree that this sentence can be improved but wish to state here that flavodoxins switch between stable SQ and rather instable HQ, which causes the negative redox potential... 'that switch between stable neutral semiquinone (SQ) and instable anionic hydroquinone (HQ) states'.

4. L92-96: E452, W459, I542, K655 and E658 are not found in Figure 3. The residues should be indicated in Figure 3 or somewhere.

Fig. 3 focuses on the negative electrostatic potential, the amino acids involved in re- and si-binding are better visualized in Supplementary Fig. 1B to which we now refer to at the appropriate place. We added the missing amino acids in the Supplementary Fig. 1B and refer now to this figure. (We also corrected the erroneously E452 to E541 in the text). Finally, we rephrased the next sentence to demonstrate that Fig. 3 aims to show the negatively charged residues (E446, D541, D649).

5. L105: The value of "the exceptionally low potential of the NCoA/DHNCOA couple" (-493 mV?) would be shown again.

Done.

6. L114: Supplementary Figure 2 is not helpful to understand that the distance to the FAD is expected to be below 14Å.

The reviewer is right. We do not refer to Fig. S2 anymore and state that the distance was a structure-based estimate.

7. L112-121: The paragraph should be divided into two paragraphs. The first part (L111-114) mentions an exogenous electron donor protein for NCR, but the later parts (L115-121) mentions the [4Fe-4S] cluster in NCR.

Done.

8. L115, "predicted": It is not clear who and when predicts the presence of Fe/S cluster in NCR.

'Predicted' is now deleted, because it is indeed not relevant here anymore. (It has been predicted in previous work based on the Fe-content, and on conserved Cys-residues)

9. L131-132: R245 and E341 should be drawn in Figure 4 or somewhere.

Done in Fig. 4.

10. L133-134: The side residue of H26 seems to face to the opposite site of FMN in Figure 4.

The reviewer is right that stating H26 points to N5 of FMN is not really correct. We intended to state here that DCR has in total less negative and more positively charged amino acids in the vicinity of the FMN, which contributes to the higher potential in comparison to the FMN in NCR. We changed to ... 'in the vicinity of FMN'

11. L153-154: Km does not always reflect binding affinity of substrates. Are there any evidences to support that the substrate-binding is much faster than the other processes of the NCoA conversion by NCR?

Taken points 11 and 12 together: we now deleted the term 'high substrate affinity' and just give the number of the previously determined low Km. We believe that scientists with a basic background in enzymology are aware that a Km of around 1 µM is generally a very low one. Generally, Km-values for CoA-ester substrates are in the 10-100 µM range; but this should not be mentioned in the text.

12. L153: "High" and "low" should be used where there is a reference.

See response to point 11.

13. L176, "10 µM": The authors mention that 20 µM of NCR was used for the experiment (L 161).

These are indeed to different experiments: during titration with redox mediators/reference electrode 20 µM were used, for other titrations without mediators only 10 µM. We clarified this by stating that experiments with 10 µM were made in setups without mediators/reference electrode,

14. L181: The equation for Nernst curve fitting should be shown.

The application of the Nernst curve is nothing really new to be shown, but we now provide a sentence in the materials and Methods section (new line 371): For fitting of Nernst curves to data points, a modified Nernst equation as described by Moffet *et al.* was used.⁴³ (reference #43 was newly added).

15. L182: Q should be explained in advance.

Done here.

16. L182-184: The fitting data using $n=1$ for the Q/SQ couple and $n=1,2,4$ for SQ/HQ couple should be shown. Readers cannot believe the best fitting without worse fittings.

A new Supplementary Fig. 5 has been prepared to unambiguously show that indeed fitting with $n=1$ can be ruled out (which would have been expected by a non-cooperative electron transfer); the theoretical best fit would be with $n=2.7$, slightly favoring $n=3$ over $n=2$.

17. Figure 6b: The color of NCRox is difficult to distinguish from that of "NCRox+NCoA".

Changed.

18. Figure 6: The label of the left bottom panel need to be corrected.

Done.

19. L206-207: The differential spectra between the presence and absence of NCoA would be helpful to understand the spectral change at 500 nm.

The requested spectrum is now integrated in the graph.

20. L220-221, "FADH \cdot reduction": In the study, the reduction of FAD is not evaluated separately. "FADH \cdot reduction" does not mean the exact FADH \cdot reduction in NCR. So the expression "FADH \cdot reduction" is somewhat confusing.

In the redox titration shown in Fig. 6D, the rise and disappearance of a blue flavin SQ at 610 nm is clearly shown. Disappearance can only be assigned by reduction of the SQ to the HQ which was at -446 mV. We have also shown that this SQ is assigned to the flavin, that does not interact with NCoA, which can only be FAD. So we are quite sure to tell that we observed reduction of FADH SQ to the HQ at -446 nm.

21. L227, supplementary Figure 7: How did the authors remove the possibility of the presence of FMNH \cdot , [4Fe-4S] $^{+}$ and FADH.

This Fig. demonstrates that upon reduction of NCR with the two-electron donor DHNCoA, only an even number of electrons can be transferred to NCR, which leaves one flavin in the SQ state. Titrations with the one-electron donor dithionite clearly indicated that FAD can be reduced stepwise to SQ and HQ, whereas for FMN the SQ state was never observed. It would be really strange if in the titrations with DHNCoA the FMN would accept two hydrides from DHNCoA only to stay in a stable SQ state. A stable SQ state would be against the generally accepted behavior of hydride donors/acceptors. (E.g. never observed for NAD).

To give a rational for our statement we added 'together with results obtained in titrations with dithionite, this finding indicates....

22. L228-236, Figure 7a: The method to evaluate the NCoA conversion is not mentioned. And, error bars of "NCoA conversion" should be shown in Figure 7a.

In line 231 we clarified this point with the statement: 'The dependence of the extent of NCoA reduction by NCR on the redox potential poised was determined by analyzing samples by ultra performance liquid chromatography (UPLC)'

The setup for this experiment was very complicated: in a single experiment the UV/vis spectrum, the redox potential and NCoA content were determined in parallel. So the numbers for NCoA represent single determinations.

23. L231-233: Did the author quantitatively validated the claim?

The 'claim' is not based on quantitative measurements but on highly plausible estimations based on our experimental findings. Full reduction of all flavins was observed at E values around -470 mV and the exact redox NCoA/DHNCOA potential has recently been determined to be -493 mV. With a Nernst curve of $n=2$, a potential of 25 mV below the E'' value equals to 27.5% reduction, which is close to observed 20% reduction.

24. Supplementary Figure 8: The figure should be shown in the main body.

Done as requested by reviewer #1.

25. L265-266: The sentence should refer to Figure 1.

Done.

26. L266-267: The sentence should refer to Figure 5 and supplementary Figure 4.

Done.

27. L279-289: The sentence should refer to Figure 5.

Done.

Reviewer #3 (Remarks to the Author):

Reduction of 2-naphtoyl-CoA to 5,6 dihydro-2-naphtoyl-CoA by 2-naphtoyl-CoA reductase requires FAD, an iron sulfur cluster and FMN as cofactors. This reaction is below the negative redox limit usually encountered in biological hydride transfer using these cofactors. The paper by Willistein et al addresses the structural and biochemical basis of this unusual flavin redox chemistry by a combination of crystallography, spectroscopy and computational studies. The authors propose a mechanism by which the enzyme generates the extremely low-potential FMNH⁻ donor necessary for the reaction to occur. The paper is an important advance in understanding enzymatic hydride transfer at the limit of redox chemistry using flavins.

Comments:

The authors should include a composite omit map or the initial map after a MR solution was obtained showing the electron density for the bound substrate, NCoA as a separate panel in figure 5.

We have added the requested composite omit map in Fig. 5 (now Fig. 5B)

Based on the PDB validation reports, there appears to be a significant problem with the crystal structure obtained from the co-crystallized enzyme-substrate complex. Not only are the B-values after refinement very high, the RSRZ value, i.e. the fit of the modelled residues to the electron density is with more than 20% outliers at the lowest end of all structures deposited in the PDB. The authors should address these problem in the manuscript.

We agree with the reviewer that the B factor of the co-crystallized enzyme-substrate/product complex is extremely high; the electron density is in some regions nearly disordered. Nevertheless, the model can be incorporated throughout the chain. The B-factor of the C-terminal region (505-625) is ca. 130 Å², of the FAD binding domain (400-505) ca. 100 Å² and of the OYE-like domain (1-400) ca. 65 Å². NCoA has a B-factor of 89.7 Å² and is nearly completely occupied. The overall architecture of NCR and the cofactor/substrate binding sites are already established by the other two structures. We included this flexible structure into the manuscript to show that substrate/product binding is not accompanied by larger conformational changes which might be prevented by the crystal lattice in the structure of NCR soaked with NCoA. We included a sentence concerning the flexibility of the co-crystallized enzyme-substrate complex.

B- values for the bound ligands, FAD, FMN and substrate should be included in table 1. It would also be helpful to add the B-factor from the Wilson plot to this table.

The B-values of FAD, FMN, the [4Fe-4S] cluster and NCoA are now integrated into Table S1. The B-factor of the Wilson plot is also listed.

There is a significant variation in the number of water molecules modelled, for instance 150 and 250 for structures at the same resolution and only 9 at a slightly lower resolution – why?

We checked the waters in NCR and the NCR-NCoA (soaked) complex structures which are based on the same crystal form. In the final refinement round, we fitted 187 solvent molecules in the NCR and 224 solvent molecules in the NCR-NCoA complex electron density. Most solvent molecules are at the same position in the two structures. Why there are still significant differences of 37 solvent molecules remains unclear. It has to be considered that at ca. 2.2 Å many water molecules just become resolved such that minor differences of the crystals, of freezing and of data collection can

explain the deviations. The very high B-factor of the structure of NCR co-crystallized with DHNCoA at 2.4 Å is a plausible reason for the low amount of solvent molecules visible.

PDB accession codes should be added to the manuscript.

The pdb codes for NCR, NCR-NCoA (soaked) and NCR-DHNCoA (co-cryst) are 6QKG, 6QKR and 6QKX, respectively. They are added to the manuscript.

In figure 2 of the ms and figure 2 in the supplementary the authors propose the binding site for the the electron donor, a hypothetical low-potential ferredoxin-like protein. How was this site identified and what would be the basis for this proposal? Figure S2 highlights the negative electrostatic potential of this binding site. Would this negative potential not counteract electron transfer from the ferredoxin to FAD?

The criteria for the proposal was to position a ferredoxin as close as possible to FAD to analyze whether the distance to its [4Fe-4S] cluster is shorter than about 14 Å. More elaborated model building attempts are not undertaken because the natural electron donor and its structure is unknown. The negative potential is essential for tuning the one-electron redox property of FAD (destabilization of FADH⁻). Potential electron transfer routes are lined by not more than one acidic residues. Therefore, no obvious barrier for electron transfer could be identified. Alternative electron donor binding sites can, of course, also not excluded.

Page 6, line 116: occupancy, not occupation

Page 19, line 364: Table 1, not Table 3

Both corrections were made.

Reviewer #4 (Remarks to the Author):

This is an interesting paper combining a range of experimental and computational studies designed to throw light on an important enzymatic mechanism. I am not an expert on the enzyme itself and only comment in depth on the computational aspects.

A standard QM/MM protocol has been followed. My major concern is the use of a B3LYP functional for the QM part. Whilst this functional generally predicts quite good structures, it often fails to predict reliable energetics, due in large part to its failure to properly model dispersive interactions which are expected to be important in bio systems. Most current QM/MM studies now include dispersive effects either explicitly by the use of the DFT-D method or implicitly by the use of modern functionals such as the M0 family of Truhlar.

I believe that the calculations should be repeated using one or more of these modern functionals to give confidence in the validity of the predictions.

We agree with the reviewers' concern that the use of B3LYP in the QM part is not optimal. As suggested we carried out recalculations using the M06 suite. We ended with an only slightly higher transition state of 15.6 kcal mol⁻¹ (rounded to 16, as the accuracy cannot be given).

Reviewers' comments:

Reviewer #1 (Remarks to the Author):

I have reviewed the point by point response letter and revised manuscript and believe that the points raised have been addressed successfully with this new version of the manuscript.

Reviewer #2 (Remarks to the Author):

The authors revised the manuscript in accordance with the reviewer's suggestions. Now the manuscript is acceptable for the publication.

Reviewer #3 (Remarks to the Author):

I have some concerns about the occupancy of the substrate in the enzyme-substrate complex. The omit map is contoured at 0.5 sigma, which is below the significance level. The problem is also indicated by the B-factor for the substrate which with 90 Å² is almost twice as high as for the bound flavins (see table S1). These observations strongly indicate that the occupancy of the substrate is significantly lower than 1. A low occupancy of the substrate could potentially not be sufficient to trigger conformational changes in the crystal lattice. In order to make readers with no crystallographic expertise aware of the problems the authors must comment on this issue.

Related to this is the bad quality of the co-crystallized enzyme-substrate structure which in fact is below standard criteria in the protein crystallography field. On the one hand the authors claim that this structure is highly similar to the one obtained by soaking, therefore providing evidence that no conformational changes occur upon substrate binding. On the other hand the authors concur that the co-crystallized structure is highly flexible and partly disordered. This means that there are significant differences in structure, indicating for example order-disorder transitions. Whether or not these are of biological significance or related to crystal lattice effects is not clear but again the non-expert reader should be made aware of these potential problems.

Other minor comments were addressed satisfactorily in the revised manuscript.

Reviewer #4 (Remarks to the Author):

The authors have addressed my concerns about the use of an inappropriate density functional

Poin to point response to reviewer suggestions in the revised version (responses shown in blue).

Reviewer #3:

I have some concerns about the occupancy of the substrate in the enzyme-substrate complex. The omit map is contoured at 0.5 sigma, which is below the significance level. The problem is also indicated by the B-factor for the substrate which with 90 Å² is almost twice as high as for the bound flavins (see table S1). These observations strongly indicate that the occupancy of the substrate is significantly lower than 1. A low occupancy of the substrate could potentially not be sufficient to trigger conformational changes in the crystal lattice. In order to make readers with no crystallographic expertise aware of the problems the authors must comment on this issue.

Related to this is the bad quality of the co-crystallized enzyme-substrate structure which in fact is below standard criteria in the protein crystallography field. On the one hand the authors claim that this structure is highly similar to the one obtained by soaking, therefore providing evidence that no conformational changes occur upon substrate binding. On the other hand the authors concur that the co-crystallized structure is highly flexible and partly disordered. This means that there are significant differences in structure, indicating for example order-disorder transitions. Whether or not these are of biological significance or related to crystal lattice effects is not clear but again the non-expert reader should be made aware of these potential problems.

We thank reviewer #3 for spotting this issue and take the concerns raised with regard to the newly added omit map together with the B-factor seriously. However, we are absolutely confident that the structural data of the naphthoyl-CoA binding is significant and substantial, and we added supporting data in the Figs below (for review).

We fully agree that the substrate is not completely occupied, but the 2F_o-F_c electron density is sufficient to identify all its components (at the sulfur is the high ED). The electron density of the soaked CoA compound is for all parts higher than 1σ (to illustrate this, we added the first Figure below). The electron density for the omit map is lower (as expected) but its profile clearly represents the extended naphthoyl-CoA (see Fig. 5b in the manuscript) and is markedly higher than the noise. The rather low σ value of 0.5 in Fig. 5b was simply chosen to avoid an interruption in the electron density. To explain this in more clearly manner we added the following sentences:

P.7., L.141: 'No significant conformational changes were found between the two binary NCR complex structures and between the substrate/product free NCR as documented in rms deviations of 0.45 Å and 0.49 Å, respectively.'

P.9 Fig. 5 legend: 'Although not completely occupied, all moieties of NCoA are also clearly visible in the 2F_o-F_c electron density map at a contour level of 1σ.'

P19 L395 (Materials and Methods): 'Although not completely occupied, all component parts of NCoA were clearly visible in the 2F_o-F_c electron density map at a contour level of 1σ. The composite omit map at a contour level of 0.5 σ is shown in Figure 5b'

We agree with the reviewer that the co-crystallized binary enzyme complex structure is in some parts highly disordered. But: the polypeptide model can be incorporated throughout the entire chain, and the R_{free} factor of 27.3 % is still on an acceptable level considering the high temperature factor of the chain and the low amount of solvent molecules fitted. As the response of the comments of reviewer 3 we already mentioned that the quality of the model is not homogeneous along the polypeptide chain. The region around the CoA compound and also the CoA compound fits properly to the

electron density. The rms deviation between soaked and co-crystallized complex structure is 0.45 Å which does not argue for significant long-range conformational changes (second Figure below). We are convinced that despite the structural deficiencies the co-crystallized binary complex is valuable. It is crystallized in a distinct crystal form with another crystal lattice and is still arranged in the same conformation as in the soaked binary complex. Both independently obtained structures together provide reasonable evidence that substrate binding is not accompanied by substantial conformational changes.

It has to be further mentioned that the B-factors of the biological relevant naphthoyl group is 65 Å² and 77 Å² in the soaked and co-crystallized binary complexes, respectively, which is not dramatically higher than the approximate values of the surrounding areas (52 Å² and 65 Å²). The CoA tail of CoA is frequently characterized by a higher temperature factor because of its exposure to bulk solvent. We thank again and hope, we can convince the reviewer that the structural data are serious despite some deficiencies and that our interpretations are justified.

Ncr-NCoA complex (soaked)
2F_oF_c map, σ=1

Ncr-NCoA complex (co-crystallized)
2F_oF_c map, σ=1

Ncr-soaked with NCoA (OYE-like domain, FAD domain, α/β domain)
Ncr-co-crystallized with DHNCoA superimposed

REVIEWERS' COMMENTS:

Reviewer #3 (Remarks to the Author):

The authors addressed my concerns satisfactorily.